# HIV-specific CD8+ T-cell proliferative response 24 weeks after early antiretroviral therapy initiation is associated with the subsequent reduction in the viral reservoir

Pien Margien van Paassen[1,2]\*[†], Alexander O Pasternak[2,3]\*[†], Dita C Bolluyt[2,3], Karel A van Dort[1,2], Ad C van Nuenen[1,2], Irma Maurer[1,2], Brigitte Boeser-Nunnink[1,2], Ninée VEJ Buchholtz[4], Tokameh Mahmoudi[5], Cynthia Lungu[5], Reinout van Crevel[6], Casper Rokx[7], Jori Symons[4], Monique Nijhuis[4], Annelou LIP van der Veen[8], Liffert Vogt[9,10], Michelle J Klouwens[8], Jan M Prins[8], Neeltje A Kootstra[2‡], Godelieve J de Bree[8‡]

[1]Department of Experimental Immunology, Amsterdam UMC, Amsterdam, Netherlands; [2]Amsterdam Institute for Infection and Immunity, Amsterdam UMC, Amsterdam, Netherlands; [3]Department of Medical Microbiology and Infection Prevention, Laboratory of Experimental Virology, Amsterdam UMC, Amsterdam, Netherlands; [4]Department of Medical Microbiology, Translational Virology, University Medical Center Utrecht, Utrecht, Netherlands; [5]Department of Biochemistry and Department of Pathology, Erasmus Medical Center, Rotterdam, Netherlands; [6]Department of Internal Medicine and Radboud Center for Infectious Diseases, Radboud University Medical Center, Nijmegen, Netherlands; [7]Department of Internal Medicine and Department of Medical Microbiology and InfectiousDiseases, Erasmus University Medical Center, Rotterdam, Netherlands; [8]Department of Internal Medicine, section Infectious Diseases, Amsterdam UMC, location AMC, Amsterdam, Netherlands; [9]Apheresis Unit,Dianet, location Amsterdam UMC, Amsterdam, Netherlands; [10]Department of Internal Medicine, section Nephrology, Amsterdam UMC, location AMC, Amsterdam, Netherlands

\*For correspondence:
p.vanpaassen@amsterdamumc.nl (PMvP);
a.o.pasternak@amsterdamumc.nl (AOP)

[†]These authors contributed equally to this work
[‡]These authors also contributed equally to this work

## eLife Assessment

The findings of this study are **valuable** as it demonstrates that when treatment is initiated during acute infection, HIV specific CD8 T cell responses are maintained long term and continued proliferative capacity of these cells may play a role in reducing HIV DNA levels. The evidence supporting the conclusions are **solid** with rigorous and advanced methodology used with the major limitations being that the findings are association level and do not meet strict criteria for causality. The work is of interest to the HIV cure field and suggests that enhancing early HIV specific CD8 T cell responses should be considered in the design of interventional cure strategies.

**Abstract** Antiretroviral therapy (ART) initiated in the acute phase of HIV infection (AHI) results in a smaller viral reservoir. However, the impact of early HIV-specific T-cell responses on long-term reservoir dynamics is less well characterized. Therefore, we measured the size of the viral reservoir

and functionality of HIV-specific CD8+ T-cell responses after the acute phase at 24 and 156 weeks after ART initiation in people with HIV who started treatment during AHI. A significant reduction in total and defective HIV DNA and a trend toward a reduction in intact HIV DNA were observed between 24 and 156 weeks. Functional CD8+ T-cell responses against HIV peptides Env, Gag, Nef, and Pol were maintained over 3 years after treatment initiation. The proliferative capacity of HIV-specific CD8+ T-cells at 24 weeks of ART was predictive of the degree of reduction in total and defective HIV DNA between 24 and 156 weeks, suggesting HIV-specific CD8+ T-cells may at least partially drive the decline of the viral reservoir. Therefore, enforcing HIV-specific immune responses as early as possible after diagnosis of AHI should be a central focus of HIV cure strategies.

## Introduction

Current treatment of HIV infection with antiretroviral therapy (ART) successfully suppresses viral replication, halts and reverses disease progression, and gives people with HIV (PWH) a near-normal life expectancy (*Trickey et al., 2017*). However, ART does not clear HIV infection and must be taken lifelong. This is due to the persistence of the viral reservoir, which remains the central barrier to achieving HIV eradication or long-term remission. The long-term persistence of viral reservoir is the result of HIV integration into the host genome: the integrated provirus persists for the lifespan of the infected cell, and if this cell divides, the provirus is passed on to its progeny. As soon as ART is interrupted, the reservoir will almost inevitably fuel prompt viral rebound (*Chun et al., 1999*; *Davey et al., 1999*; *Rothenberger et al., 2015*).

During the early stages of HIV infection, also referred to as acute HIV infection (AHI), active viral replication supports rapid seeding of the viral reservoir within lymph nodes and other tissues like the spleen and gut-associated lymphoid tissue (*Daar et al., 1991*; *Pierson et al., 2000*). HIV reservoir is established extremely early after infection: a study in nonhuman primates (NHPs) has shown that this happens even before detectable viremia (*Whitney et al., 2014*). The early immune response during AHI is critical in shaping the course of infection and the size of the viral reservoir (*Whitney et al., 2014*). Along with rising viral levels, cytokine and chemokine levels increase, which recruit and activate innate immune cells (*Stacey et al., 2009*; *Altfeld and Gale, 2015*). These regulate the subsequent adaptive immune response (*Borrow, 2011*). HIV-specific CD8+ cytotoxic T lymphocytes (CTLs) appear 2–3 weeks after infection and are thought to contribute to the initial decline in plasma viremia before ART is initiated (*Borrow et al., 1997*; *Koup et al., 1994*). Conversely, active viral replication shapes the magnitude and diversity of HIV-specific CD8+ T-cells, especially during this early stage of infection (*Takata et al., 2017*). Moreover, CD8+ T-cells exert immune pressure by recognizing and killing infected cells that present HIV-derived peptides via HLA class I molecules. Indeed, individuals expressing protective HLA alleles such as HLA-B57 and HLA-B27 tend to mount more effective CD8+ T-cell responses and exhibit lower viral set points (*Migueles et al., 2000*).

In addition to limiting active viral replication, CD8+ T-cells may also influence the composition and dynamics of the viral reservoir. Although ART effectively halts new rounds of infection, it does not eliminate long-lived infected cells harboring intact proviral DNA. Emerging evidence suggests that CD8+ T-cell pressure can shape which proviruses persist by selectively eliminating infected cells that express viral antigens during latency reversal or low-level transcription (*Einkauf et al., 2022*; *Ho et al., 2013*).

The importance of the host CD8+ T-cell response in controlling virological outcomes is supported by many studies. For instance, strong HIV-specific CD8+ T-cell responses have been shown to control HIV replication without ART in human elite (*Migueles et al., 2002*; *Migueles et al., 2008*; *Migueles et al., 2009*; *Boppana and Goepfert, 2018*) and posttreatment controllers (*van Paassen et al., 2023*; *Blazkova et al., 2021*), and to control SIV replication in NHPs (*Chowdhury et al., 2015*; *McBrien et al., 2020*; *Passaes et al., 2024*). The quality of the CD8+ T-cell response is defined by its breadth, magnitude, polyfunctionality, and cytolytic capacity and has been associated with better viral control and slower disease progression (*Migueles and Connors, 2010*; *Betts et al., 2006*; *Ndhlovu et al., 2015*). However, in the majority of PWH, CD8+ T-cell dysfunctionality is already observed shortly after peak viremia during AHI, which could potentially be prevented if ART is initiated prior to this occurring (*Takata et al., 2017*; *Ndhlovu et al., 2019*). Indeed, effective HIV-specific CD8+ T-cell responses were observed in people treated during AHI, and the magnitude of this response correlated with the

transcriptional activity of the virus (*Takata et al., 2023*). ART initiation during AHI also resulted in a smaller viral reservoir size by limiting viral replication and seeding of the reservoir compared to treatment initiation during chronic infection (CHI) (*Koup et al., 1994*; *Takata et al., 2017*; *Ananworanich et al., 2016*; *Jain et al., 2013*; *Archin et al., 2012*; *Strain et al., 2005*; *Ananworanich et al., 2012*). Moreover, early ART initiation enhances restoration of the immune system (*Ananworanich et al., 2012*; *Le et al., 2013*). The importance of ART initiation during AHI is underscored by the fact that early treated individuals have a higher chance of achieving at least temporary viral control after stopping ART (*Namazi et al., 2018*).

Currently, it remains unknown whether early ART can preserve functionality of HIV-specific CD8+ T-cell responses and how these relate to the viral reservoir long term. In this study, we aimed to characterize these dynamics in participants of the Netherlands Cohort Study on Acute HIV Infection (NOVA study), who initiated ART immediately after diagnosis of AHI and were followed for over 3 years. We found evidence that the early immune response shapes aspects of the viral reservoir when ART is initiated during AHI.

**Table 1.** Characteristics of the study participants*.

| | | Total (n=22) | Paired analysis (n=12)[†] |
|---|---|---|---|
| Sex at birth, male | | 100% | |
| Age at inclusion, years[‡] | | 38 (28–48) | 40.5 (25.5–49) |
| Country of birth | The Netherlands | 18 | 12 |
| | Other | 4 | 0 |
| Fiebig stage at diagnosis[§] | II | 1 | 0 |
| | III – IV | 12 | 7 |
| | V | 5 | 4 |
| | VI | 4 | 1 |
| Plasma viral load at baseline, $10^6$ copies/mL | | 0.4 (0.04–10) | 0.61 (0.28–10) |
| CD4+ T-cell count at baseline, cells/mm$^3$ | | 445 (308–593) | 435 (295–555) |
| HIV subtype | B | 15 | 10 |
| | CRF02_AG | 1 | 0 |
| | AE | 3 | 1 |
| | F | 2 | 1 |
| | Unknown | 1 | 0 |
| Time between diagnosis and start of ART | Same day | 15 | 8 |
| | 1 day | 3 | 1 |
| | 2 days | 2 | 1 |
| | Unknown | 2 | 2 |

*Numerical data provided in **Table 1—source data 1**.

[†]Participants with available samples from both 24-week and 156-week time points.

[‡]Medians and interquartile ranges are shown for continuous variables.

[§]Fiebig, E. W. et al. Dynamics of HIV viremia and antibody seroconversion in plasma donors: implications for diagnosis and staging of primary HIV infection. AIDS 17, 1871–1879 (2003).

The online version of this article includes the following source data for table 1:

**Source data 1.** Numerical data corresponding to *Table 1*.

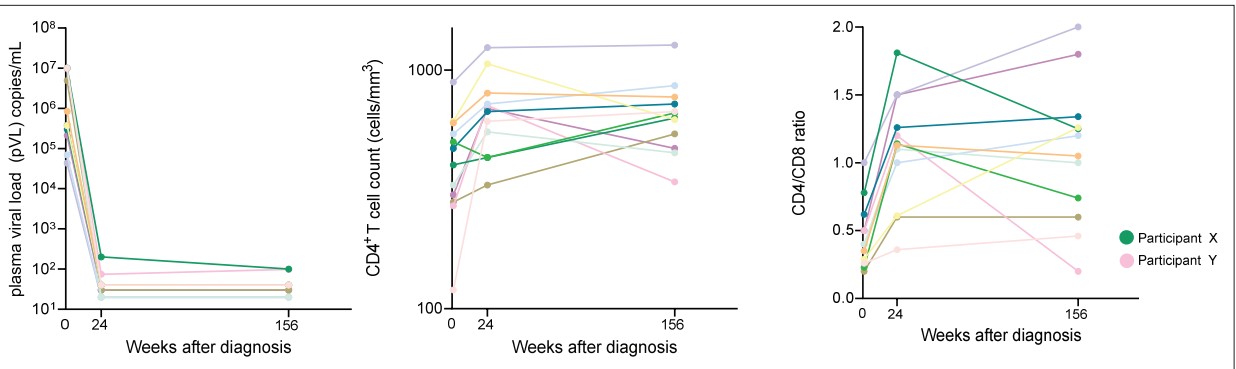

**Figure 1.** Plasma viral load, CD4+ T-cell count, and CD4/CD8 ratio at baseline, 24 and 156 weeks post-antiretroviral therapy (ART) initiation. Participant X (green dot) had detectable loads of 200 and 100 copies/mL at 24 and 156 weeks. Participant Y (pink dot) had detectable loads of 74 and 98 copies/mL at 24 and 156 weeks. Lower limit of quantification of the assays used ranged between 40 copies (earlier) and 20 copies (later time points) (see Methods). Numerical data provided in *Figure 1—source data 1*.

The online version of this article includes the following source data for figure 1:

**Source data 1.** Numerical data corresponding to *Figure 1*.

# Results

## Cohort description

This study included 22 participants of the NOVA cohort who initiated ART immediately (median 1 day, IQR 0–2) after diagnosis during AHI. We selected those participants for whom leukapheresis samples at either 24 weeks and/or 156 weeks post-ART initiation were available. Samples from both time points were available from 12 of 22 participants, resulting in a total of 34 samples. Baseline characteristics are provided in *Table 1*. The median age at time of inclusion was 38 years (IQR 28–48), all participants were male, and two-thirds of participants had a subtype B HIV infection. At diagnosis, participants had a median plasma viral load (pVL) of $0.4 \times 10^6$ copies/mL (*Figure 1*). Two participants had a detectable pVL of 200 and 100 copies/mL (Participant X) and 74 and 98 copies/mL (Participant Y) at 24 and 156 weeks, respectively. CD4+ T-cell counts at 24 and 156 weeks were comparable, with a median of 610 (IQR 520–725) and 695 (IQR 623–838) cells/mm$^3$, respectively (*Figure 1*; p=0.13), as were the CD8+ T-cell counts (median 680 [IQR 485–840] at 24 weeks and 730 [IQR 520–1005] cells/mm$^3$ at 156 weeks [p=0.41]). The CD4/CD8 ratio increased significantly between baseline and 24 weeks (*Figure 1*; p<0.01), but was comparable between 24 and 156 weeks.

## Longitudinal reduction in total and relative increase in transcription-competent viral reservoir

First, we determined the HIV reservoir size at 24 and 156 weeks by measuring total, intact, and defective HIV DNA in PBMCs. Intact and defective proviruses were measured by intact proviral DNA assay (*Bruner et al., 2019*). Intact (*psi+ env+*) proviruses were detected in 33 of 34 samples, while defective proviruses (either 3' defective [*psi+ env-*] or 5' defective [*psi- env+*]) were detectable in all samples. A number of studies have shown that viral latency does not preclude viral transcription (*Fombellida-Lopez et al., 2024*). Therefore, we also quantified cell-associated unspliced (US) HIV RNA as a measure of transcriptionally active reservoir at 24 and 156 weeks. To better understand the relationships between these different measures of the viral reservoir, Spearman correlation analysis was conducted at 24 and 156 weeks. At both time points, strong positive correlations were observed between the levels of total HIV DNA, 3' defective HIV DNA, and US RNA (*Figure 2*). At 24 weeks but not at 156 weeks, nonsignificant trends toward positive correlations of intact HIV DNA with total DNA and US RNA were observed (p=0.09 and p=0.10, respectively), and both total DNA and US RNA did not correlate with 5' defective DNA at any time point (*Figure 2*).

At the 24-week time point, replicating virus could be isolated by quantitative viral outgrowth assay (QVOA) from CD4+ T-cells of six participants, one of whom had a detectable pVL at that time point (Participant Y, 74 copies/mL). For four out of these six participants, 156-week samples were available, but replication-competent virus could not be retrieved from any of them. We compared the

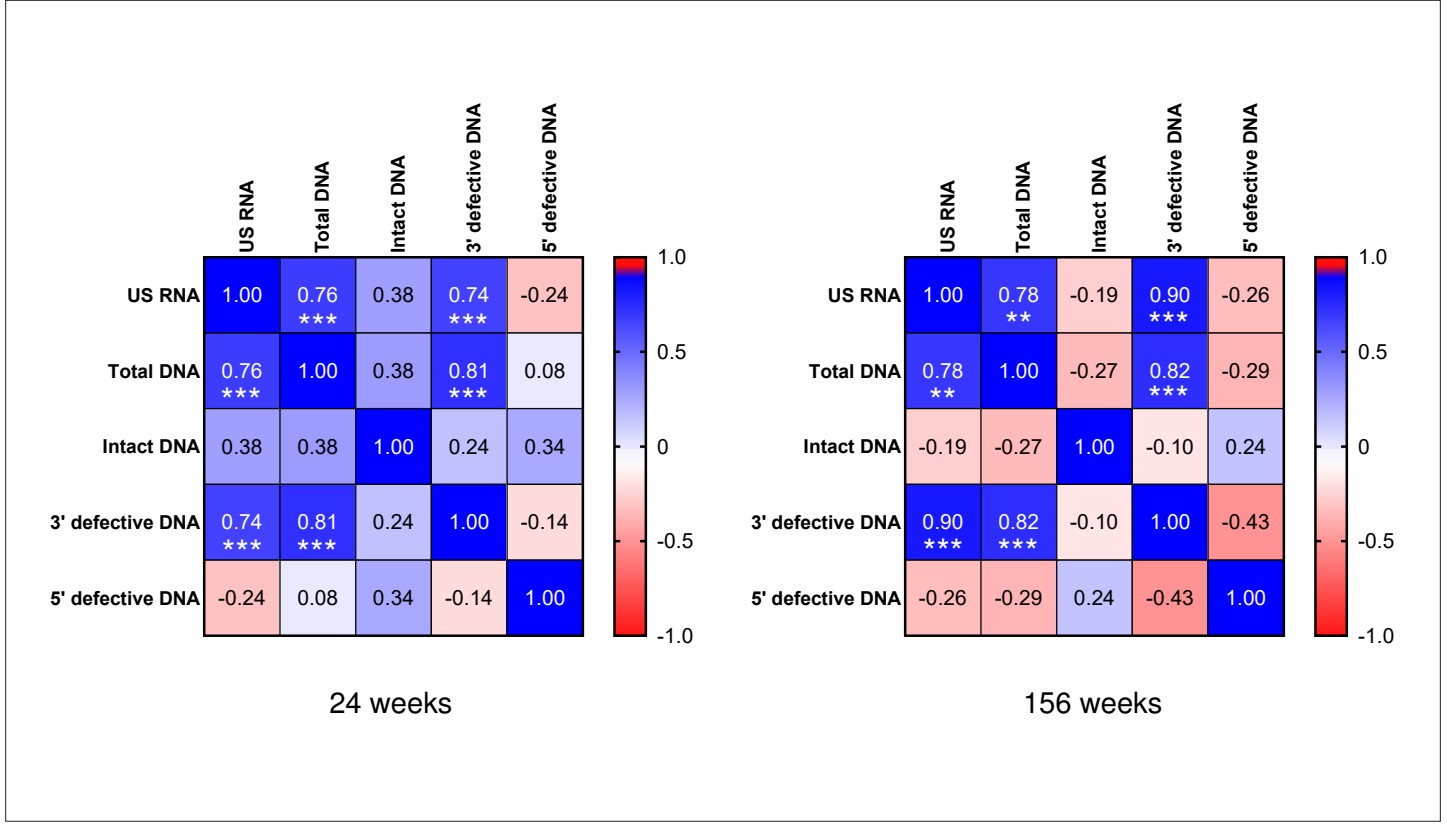

**Figure 2.** Correlation matrix of viral reservoir measures at 24 and 156 weeks after start of antiretroviral therapy (ART). Correlation coefficients (*rho*) determined by Spearman correlation analyses are shown. Strength of positive and negative correlations is indicated by the color shade displayed in the legend. Significant correlations are indicated by \*\*\* if p<0.001 or by \*\* if p<0.01. All significant correlations remained significant after corrections for multiple comparisons. Numerical data provided in *Figure 2—source data 1*.

The online version of this article includes the following source data and figure supplement(s) for figure 2:

**Source data 1.** Numerical data corresponding to *Figure 2*.

**Figure supplement 1.** Comparison of intact HIV DNA levels at 24 weeks antiretroviral therapy (ART) between participants with positive (n=6) and negative (n=14) quantitative viral outgrowth assay (QVOA).

**Figure supplement 1—source data 1.** Numerical data corresponding to *Figure 2—figure supplement 1*.

intact HIV DNA levels at the 24-week time point between the six participants, from whom we were able to isolate replicating virus, and the fourteen participants, from whom we could not. Participants with positive QVOA had significantly higher intact HIV DNA levels than those with negative QVOA (p=0.029, Mann-Whitney test; *Figure 2—figure supplement 1*). Five of six participants with positive QVOA had intact DNA levels above 100 copies/$10^6$ PBMC, while thirteen of fourteen participants with negative QVOA had intact HIV DNA below 100 copies/$10^6$ PBMC (p=0.0022, Fisher's exact test). These findings indicate that recovery of replication-competent virus by QVOA is more likely in individuals with higher levels of intact HIV DNA in intact proviral DNA assay (IPDA), reaffirming a link between the two measurements.

Next, we included participants with available samples from both time points (n=12) in a comparative analysis of the reservoir measures between 24 and 156 weeks ART (*Figure 3*). Significant reductions in levels of total (p=0.02), 3' defective (p<0.01), and total defective (p<0.01) HIV DNA were observed from 24 to 156 weeks. Furthermore, we noted a trend toward a reduction in intact HIV DNA level from 24 to 156 weeks (p=0.11). No significant differences in 5' defective HIV DNA, as well as US RNA levels (p=0.23 for both comparisons), were observed between 24 and 156 weeks. However, relative HIV transcription level per provirus, calculated as US RNA/total HIV DNA (US/TD) ratio, significantly increased between 24 and 156 weeks (p=0.03), indicating a relative increase in transcriptional activity of the reservoir over time.

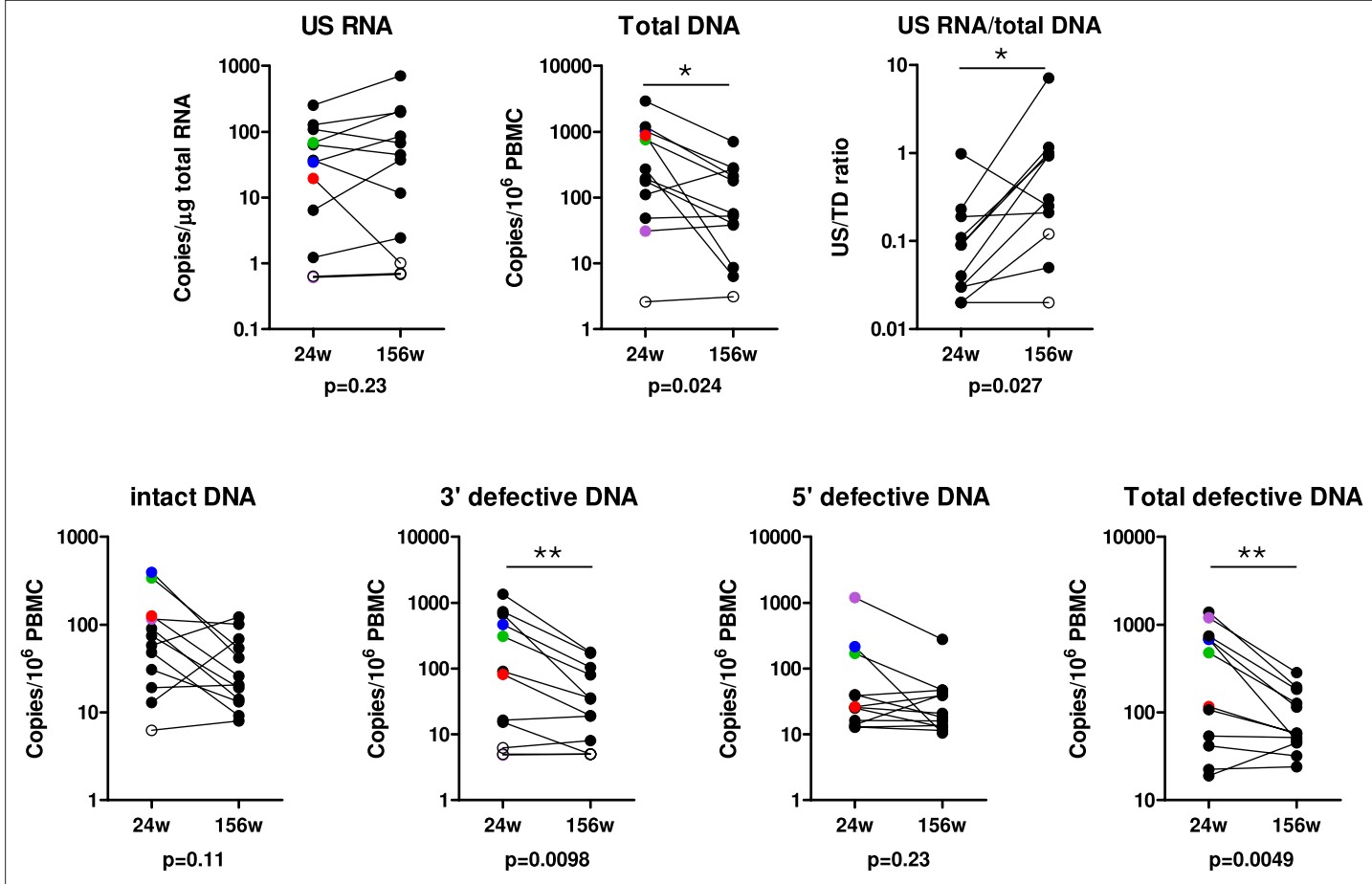

**Figure 3.** Paired comparisons of viral reservoir measures at 24 and 156 weeks of antiretroviral therapy (ART). Open symbols represent values below the detection limits of the assays. The four participants with a positive quantitative viral outgrowth assay (QVOA) at 24 weeks are marked with colors (purple, blue, green, or red). A paired Wilcoxon test was performed to test the significance of the differences between the time points (* if p<0.05 or ** if p<0.01). Pairs where both values were undetectable, or where one was undetectable and its detection limit was higher than the value of the detectable partner, were excluded from the analysis. Exact p-values are indicated below the graphs. Numerical data provided in *Figure 3—source data 1*.

The online version of this article includes the following source data for figure 3:

**Source data 1.** Numerical data corresponding to *Figure 3*.

## Comparable frequency and function of the HIV-specific CD8+ T-cell response at 24 and 156 weeks

General phenotyping of CD4+ and CD8+ T-cells showed no difference in frequencies of naïve, memory, or effector CD4+ and CD8+ T-cells between 24 and 156 weeks (*Figure 4A*). Moreover, CD8+ T-cell activation levels were low (<10%) and remained stable over time. Next, HIV-specific CD8+ T-cell numbers and functionality at 24 and 156 weeks post-ART initiation were analyzed. A subgroup of participants (n=9), positive for HLA-type HLA-A*2, HLA-B*7, or both, showed similar frequencies of HIV-specific dextramer positive CD8+ T-cells at 24 and 156 weeks (median frequency 0.066% [IQR 0.031–0.11] at 24 weeks and 0.055% [IQR 0.052–0.11] at 156 weeks) (*Figure 4B*, left panel; p=0.48). The phenotype of HIV-dextramer-specific CD8+ T-cells showed no difference in expression of exhaustion markers (upregulation of PD-1, CTLA-4, and CD160 expression; loss of CD28 expression) between the two time points (*Figure 4B*, right panel).

HIV-specific CD8+ T-cell functionality was assessed through stimulation with HIV Env, Gag, Nef, and Pol peptide pools. The readout of these stimulations was the interferon gamma release assay (IGRA), activation-induced marker (AIM) assay, and cell proliferation (precursor frequency and proliferated cells). IFN-γ responses to Env, Gag, Nef, and Pol were observed in 3, 8, 2, and 1 participant(s), respectively, at 24 weeks and 1, 7, 2, and 1 participant(s), respectively, at 156 weeks. When frequencies of

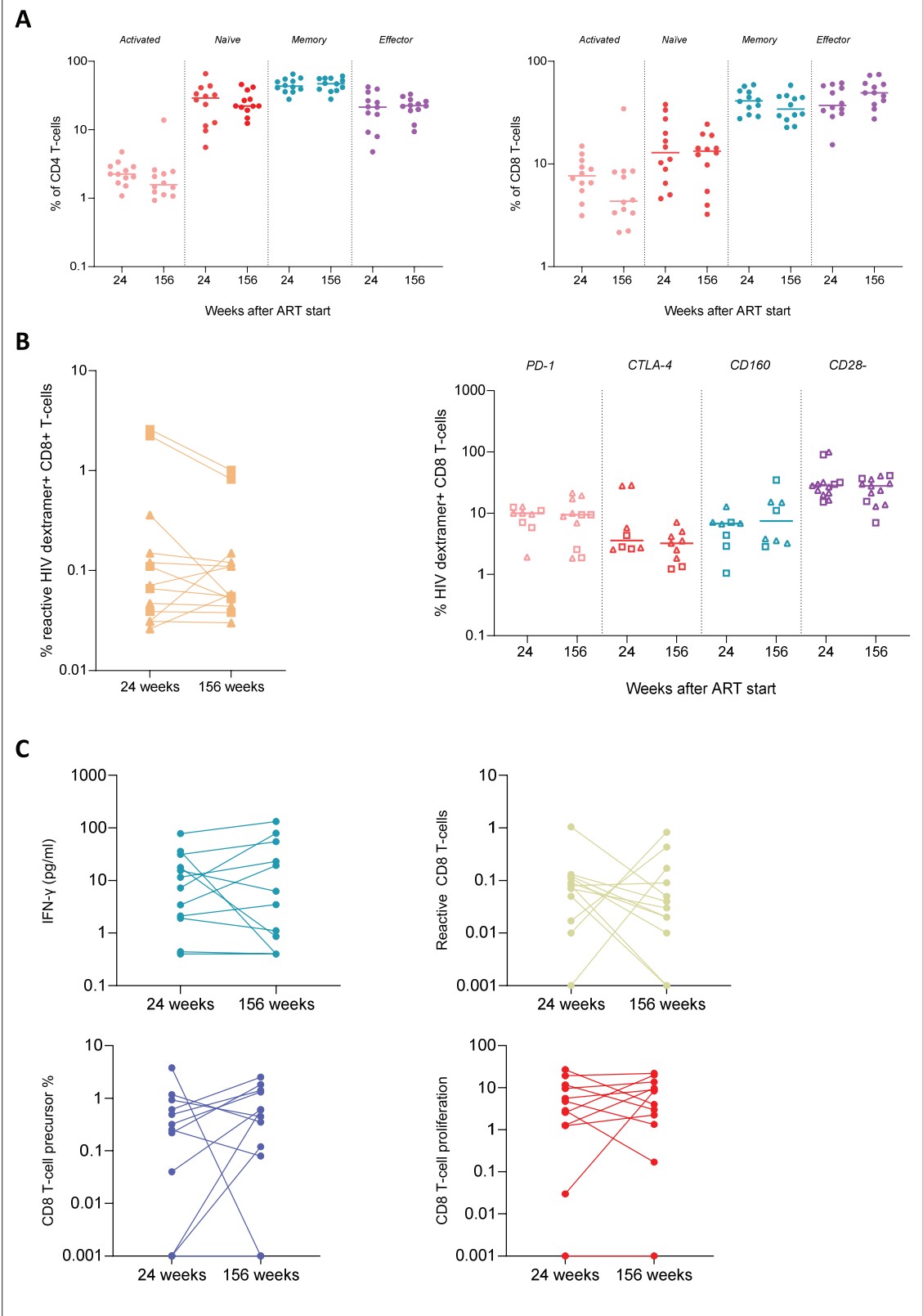

**Figure 4.** Longitudinal analysis of immunological parameters. (**A**) Longitudinal analysis of frequencies of activated and naive, memory and effector subsets within the CD4+ and CD8+ T-cell populations. Wilcoxon signed-rank test (p<0.05) was used to determine statistical significance of the differences between the time points. Numerical data provided in ***Figure 4—source data 1***. (**B**) Left panel: the frequency of dextramer+CD8+ T-cells at 24 and 156 weeks after antiretroviral therapy (ART) initiation. Right panel: the expression of exhaustion markers (upregulation of PD-1, CTLA-4, and

*Figure 4 continued on next page*

*Figure 4 continued*

CD160 expression; loss of CD28 expression) on HIV-dextramer-specific CD8+ T-cells at 24 and 156 weeks after ART initiation. For participants that were HLA-A*2 and HLA-B*7 positive, HLA-A*2 and HLA-B*7 dextramer staining was included separately. HLA-A*2 dextramer is marked as a triangle and HLA-B*7 dextramer as a square in both panels. Numerical data provided in *Figure 4—source data 2*. (**C**) HIV-specific T-cell responses upon HIV-peptide stimulation (Env, Gag, Nef, Pol) at 24 and 156 weeks after ART: interferon gamma release assay (IGRA)/IFN-γ release, activation-induced marker (AIM) reactive CD8+ T-cells, precursor frequency, and the proportion of proliferating CD8+ T-cells. Frequencies of proliferating cells in response to Env, Gag, Nef, and Pol peptides were combined. Numerical data provided in *Figure 4—source data 3*.

The online version of this article includes the following source data for figure 4:

**Source data 1.** Numerical data corresponding to *Figure 4A*.

**Source data 2.** Numerical data corresponding to *Figure 4B*.

**Source data 3.** Numerical data corresponding to *Figure 4C*.

---

proliferating cells in response to Env, Gag, Nef, and Pol peptides were combined, no significant difference over time was observed (*Figure 4C*). The AIM assay showed a similar broad HIV-specific T-cell response to at least three different viral proteins in the majority of individuals at both time points. The magnitude of the T-cell response, by combining the frequencies of reactive CD8+ T-cells to all viral proteins (Env, Gag, Nef, Pol) tested, showed no statistically significant differences (*Figure 4C*) over time. Similarly, a broad HIV-specific CD8+ T-cell proliferative response to at least three different viral proteins was observed in the majority of individuals at both time points. At 24 weeks, 6/11 individuals had a response to Env, 10/11 to Gag, 5/11 to Nef, and 4/11 to Pol. At 156 weeks, 8/11 individuals had a response to Env, 10/11 to Gag, 8/11 to Nef, and 9/11 to Pol, with no significant differences in precursor frequencies and proliferative capacity between week 24 and week 156 (*Figure 4C*).

## Magnitude of proliferative HIV-specific CD8+ T-cell response predicts the reduction in the viral reservoir

Next, we performed a Spearman correlation analysis between these responses and viral reservoir measurements. The frequencies of HIV-specific dextramer positive CD8+ T-cells did not correlate with any of the reservoir measurements at 24 or 156 weeks (*Figure 5—figure supplement 1*). Also, neither HIV-specific CD8+ T-cell functionality as determined by IGRA nor proportion of HIV-reactive (AIM) CD8+ T-cells showed any correlation with the viral reservoir at 24 weeks (*Figure 5*). However, at this time point, HIV-specific CD8+ T-cell proliferative response positively correlated with the levels of total and total defective HIV DNA (rho = 0.62, p=0.037 and rho = 0.70, p=0.014, respectively). At 156 weeks, however, these correlations were no longer observed (*Figure 5*).

To determine whether any immunological parameters measured at 24 weeks' ART were predictive for the subsequent change in the viral reservoir size, we fitted statistical models, in which we included age and ART regimen, in addition to three different measures of HIV-specific CD8+ T-cell responses, as explanatory variables, and changes in total, intact, and total defective HIV DNA between 24 and 156 weeks' ART as dependent variables (*Table 2*). As the vast majority of participants were treated with integrase strand transfer inhibitor-based ART, we only included the nucleotide reverse transcriptase inhibitor backbone as ART regimen in the models. Neither age nor ART regimen nor IFN-γ release nor CD8+ T-cell reactivity was predictive for the subsequent change in the reservoir size. However, CD8+ T-cell proliferative response at 24 weeks was predictive for the degree of reduction in total and total defective HIV DNA (p=0.014 and p=0.0017, respectively) (*Table 2*). This suggests that the early presence of HIV-specific CD8+ T-cells with an enhanced proliferative capacity in response to HIV plays a role in the reduction in the viral reservoir over the course of 2.5 years.

## Discussion

In this study, we investigated the longitudinal dynamics of the HIV reservoir and host immunological responses in people immediately treated with ART during AHI. We found a reduction in the viral reservoir, as evidenced by the significant reduction in total and defective HIV DNA and the trend toward a reduction in intact HIV DNA, between 24 and 156 weeks on ART. Strikingly, this reduction in total and defective HIV DNA levels over time was predicted by HIV-specific proliferative CD8+ T-cell responses

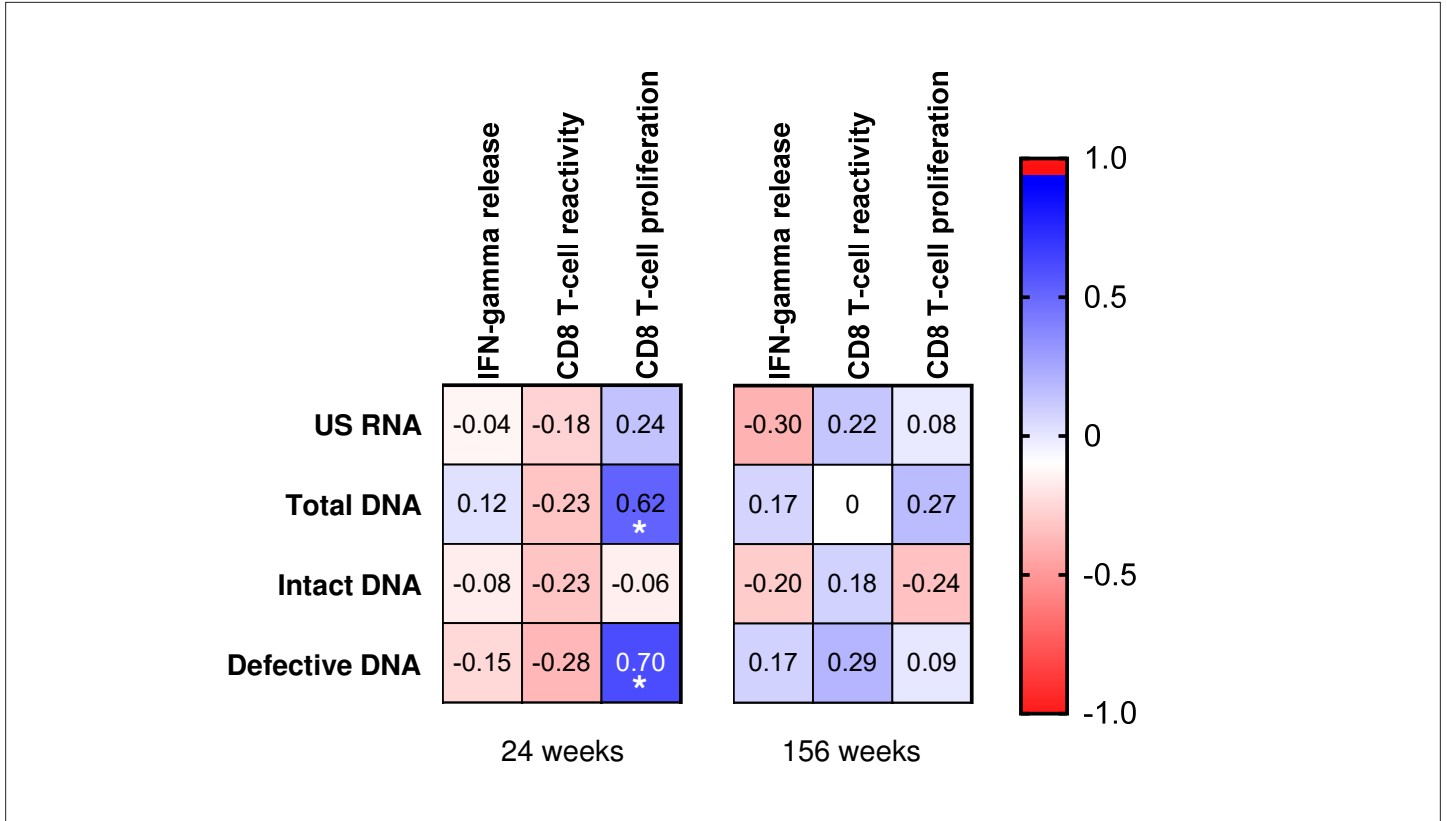

**Figure 5.** Correlations of reservoir measures and HIV-specific CD8+ T-cell responses as determined by IFN-γ release assay, activation-induced marker (AIM), and proliferation assay (proliferating cells and precursor cells). The immune responses are defined as the sum of the responses to Env, Gag, Nef, and Pol combined at 24 and 156 weeks. Correlation coefficients (*rho*) determined by Spearman correlation analyses are shown. Strength of positive and negative correlations is indicated by the color shade displayed in the legend. Significant correlations are indicated by *** if p<0.001 or by ** if p<0.01. All significant correlations remained significant after corrections for multiple comparisons. Numerical data provided in *Figure 5—source data 1*.

The online version of this article includes the following source data and figure supplement(s) for figure 5:

**Source data 1.** Numerical data corresponding to *Figure 5*.

**Figure supplement 1.** Correlation matrix of reservoir measurements and the frequency of dextramer+CD8+ T-cell responses at 24 and 156 weeks after antiretroviral therapy (ART) initiation.

**Figure supplement 1—source data 1.** Numerical data corresponding to *Figure 5—figure supplement 1*.

against HIV peptides Env, Gag, Nef, and Pol at 24 weeks. We also observed that HIV-specific CD8+ T-cell responses were maintained over 3 years after treatment initiation.

We observed that the defective HIV DNA levels decreased significantly between 24 and 156 weeks of ART. This is different from studies in CHI, where no significant decrease during the first 7 years of ART (*Peluso et al., 2020*; *Gandhi et al., 2021*), or only a significant decrease during the first 8 weeks on ART, but not in the 8 years thereafter, was observed (*Nühn et al., 2025*). The integrated but defective proviruses do not produce replicating virus but can be transcriptionally and translationally competent and are therefore thought to play a substantial role in ongoing immune activation (*Singh et al., 2023*). Indeed, it was shown that these defective proviruses are capable of producing viral RNA transcripts and proteins both in vivo and in vitro (*Imamichi et al., 2016*; *Imamichi et al., 2020*). In our cohort, at 24 weeks US RNA correlated with defective HIV DNA levels but not with intact HIV DNA. This suggests that the US RNA transcripts are mainly produced from defective proviruses. Moreover, the correlation between the HIV-specific CD8+ T-cell response and defective HIV DNA levels suggests detection by the immune system, leading to decay and shaping of the proviral landscape. This is in line with a study showing that CTLs indeed target defective proviruses (*Pollack et al., 2017*). Importantly, reservoir decay patterns are not only influenced by HIV-specific immune responses, but are also known to be associated with other factors. A recent study found that there was a faster decay of

**Table 2.** Variables measured at 24 weeks' antiretroviral therapy (ART) associated with the changes in HIV reservoir markers between 24 and 156 weeks of ART*.

| Variable | Total HIV DNA[†] | | Intact HIV DNA | | Defective HIV DNA | |
|---|---|---|---|---|---|---|
| | Effect size (95% CI) | p[‡] | Effect size (95% CI) | p | Effect size (95% CI) | p |
| Age, per year | −2.99 (−28.5 to 22.5) | 0.60 | −0.49 (−5.00 to 4.02) | 0.86 | 1.19 (−14.3 to 16.7) | 0.66 |
| NRTI backbone, ABC+3 TC vs. FTC +TDF/TAF[§] | −1.10 (−4.80 to 2.60) | 0.56 | −0.67 (−4.56 to 3.22) | 0.74 | 2.67 (−0.94 to 6.27) | 0.15 |
| IFN-γ release, per pg/mL | 2.40 (−13.2 to 18.0) | 0.79 | −1.32 (−4.04 to 1.40) | 0.59 | −5.29 (−14.4 to 3.81) | 0.14 |
| CD8+ T-cell reactivity, per % | −537 (−1677 to 603) | 0.27 | −85.6 (−298 to 127) | 0.25 | −361 (−1073 to 351) | 0.15 |
| CD8+ T-cell proliferation, per % | 62.7 (28.7–96.7) | 0.014 | 2.84 (−4.62 to 10.3) | 0.56 | 37.8 (18.5–57.1) | 0.0017 |

*Numerical data provided in **Table 2—source data 1**.
[†]Total, intact, and defective HIV DNA are in copies/$10^6$ PBMC.
[‡]Calculated by fitting generalized linear models (GLM), type III tests were used.
[§]Six participants were treated with ABC+3TC and six participants – with FTC+TDF or FTC+TAF as an NRTI backbone.

The online version of this article includes the following source data for table 2:

**Source data 1.** Numerical data corresponding to **Table 2**.

intact and defective HIV DNA when ART was initiated earlier, initial CD4+ T-cell counts were higher, and pre-ART pVL was lower (**Barbehenn et al., 2024**). Indeed, a recent study investigated reservoir dynamics in people that initiated ART during hyperacute HIV in subtype C infection and found that early ART was associated with reduced phylogenetic diversity and rapid decay of intact proviruses (**Reddy et al., 2024**). In fact, no intact provirus could be detected after 1 year of ART (reduction of 51% per month), while a decline in defective provirus was observed (reduction of 35% per month) (**Reddy et al., 2024**). In our cohort, we observed a significant decrease in total and defective HIV DNA load between 24 weeks and 3 years, while the decline of intact HIV DNA was less pronounced. This discrepancy could possibly be explained by the time period after AHI that was analyzed, as in that study, decay of intact HIV DNA was observed in the first 6 months of treatment (**Reddy et al., 2024**), while we determined the reduction between weeks 24 and 156. Similarly, in another study, a comparable biphasic decay was found in total (half-life 12.6 weeks) and integrated (half-life 9.3 weeks) HIV DNA in AHI, but total HIV DNA continued to decline in the second decay phase (**Massanella et al., 2021**). Recent mathematical modeling of reservoir decay in AHI between 0 and 24 weeks after ART initiation showed a biphasic decay for both intact and defective DNA (**Barbehenn et al., 2024**). Intact DNA showed a rapid initial $t_{1/2}$ during the first 5 weeks of ART, followed by a slower decay with a $t_{1/2}$ of around 15 weeks. Defective DNA showed an even significantly larger decrease in the first phase than intact DNA, followed by a slower decay (**Barbehenn et al., 2024**). The lack of significant reduction in intact HIV DNA in our study may also be explained by ongoing immune-mediated selection of integrated intact proviral DNA in repressive and heterochromatin locations, eventually resulting in a shift toward a state of 'deep latency', which has been suggested previously (**Lian et al., 2023**) and also seen in people who naturally control HIV (elite controllers) (**Jiang et al., 2020**; **Lian et al., 2021**). This way, these intact proviruses are not expressed and hence not eliminated by the immune system, which could explain why we do not see a significant reduction. Interestingly, we could only retrieve replication-competent virus from 6 out of 20 participants at week 24 and from none at week 156, indeed suggesting selection for integrated intact proviruses that are not rebound-competent upon reactivation.

The relationship between HIV reservoir reduction and HIV-specific CD8+ T-cell responses has been recently investigated by **Takata et al., 2023** in a study that included two Thai cohorts with participants starting ART during AHI and CHI, respectively. In CHI, they found a similar reduction in HIV reservoir over 2 years on ART; however, CD8+ T-cell responses also declined. In AHI, overall the reservoir was lower, and the frequency of reactive CD8+ T-cells 96 weeks after ART initiation was comparable to the frequency in CHI. Importantly, a larger HIV reservoir size hampered differentiation into functional HIV-specific CD8+ T-cells (**Takata et al., 2023**). In our study, we did not see a change in the number of HIV-specific CD8+ T-cell responses over time, and even more so, we found a preserved functionality up to 3 years after ART start. In contrast to the Takata study (**Takata et al., 2023**), in our cohort,

we did not observe a significant correlation between cell-associated HIV US RNA and HIV-specific CD8+ T-cell responses. Further studies should investigate whether viral transcription drives the CD8+ T-cell response. Interestingly, we observed a significant increase in the US RNA/total HIV DNA ratio between 24 and 156 weeks, suggesting a paradoxical shift toward a more transcriptionally active reservoir despite an overall reduction in the reservoir size. The mechanism behind this effect remains unclear but may involve preferential survival of transcriptionally active proviral clones with time on ART.

The use of different readouts may explain at least some of the differences observed between studies. The study by *Takata et al., 2023* reported a decline in functional CD8+ T-cell responses after 2 years of ART based on a combined AIM/intracellular staining assay (ICS) and using a different activation marker (4-1BB). In our cohort, we used both the AIM and proliferation assay to determine that HIV-specific CD8+ T-cell responses were maintained. Interestingly, relations similar to our study between the HIV reservoir size decline and HIV-specific CD8+ T-cell responses were reported (*Takata et al., 2023*). However, Takata et al. showed loss of the association when total HIV DNA was used as a reservoir marker, whereas we report here a predictive value of HIV-specific CD8+ T-cell responses for the reduction in both total and defective HIV DNA.

There are some limitations to this study. First, our cohort consists of only males, who are mostly of European descent. Therefore, our findings might not apply to people of different ethnicities or females. It is known that host genetic factors (related to different ethnicities or sex at birth) influence immune responses and viral reservoir characteristics (*Naranbhai and Carrington, 2017*). Second, not all participants underwent leukapheresis at the 156-week time point due to personal or logistic (COVID-19 pandemic) reasons, and therefore, we had a small longitudinal sample size that included participants ranging from Fiebig II-VI. Therefore, we could not assess the role of the Fiebig stage at ART initiation. A strength of our study is the long-term sampling, which allowed us to assess the reservoir decay and host immune responses years after ART initiation. Previously mentioned studies into reservoir decay have mainly reported on the first year after treatment was started (*Barbehenn et al., 2024*; *Reddy et al., 2024*; *Gunst et al., 2022*).

In summary, our study shows that between 24 weeks and 3 years of ART, total and defective HIV DNA reduced significantly and that this reduction is predicted by the magnitude of HIV-specific CD8+ T-cell responses at 24 weeks. This suggests that HIV-specific CD8+ T-cells may at least partially drive the decline of the viral reservoir. Our study has several implications. First, it confirms the complexity of host-virus interplay, as we show that defective HIV DNA decreased stronger than intact HIV DNA, and defective, but not intact, HIV DNA correlated with US RNA and the functionality of the HIV-specific CD8+ T-cell response. However, the reduction in defective HIV DNA did not result in a decreased HIV transcriptional activity over time. This could be the result of selection for cells that harbor transcriptionally active proviruses that circumvent immune surveillance through, for instance, upregulation of immune inhibitory molecules like PD-1, CTLA-4, and TIGIT, downregulation of HLA class I molecules, or the emergence of viral escape mutations. Second, our study shows that even in acute treated HIV infection, the reservoir is readily detectable despite immediate ART. We believe our study underscores that in line with what was shown in natural HIV control, an (early) functional CD8+ T-cell response is shaping the viral reservoir during ART and that enhancing host immune responses should be a focus for interventions aimed at a functional HIV cure.

## Methods

**Key resources table**

| Reagent type (species) or resource | Designation | Source or reference | Identifiers | Additional information |
|---|---|---|---|---|
| Antibody | Anti-CD4-FITC (Mouse monoclonal) | BD Biosciences | Cat# 345768; RRID:AB_2868797 | 1:100 |
| Antibody | Anti-OX40-PE (Mouse monoclonal) | BioLegend | Cat# 350004; RRID:AB_10645478 | 1:20 |
| Antibody | Anti-CD69-PE-Cy7 (Mouse monoclonal) | BioLegend | Cat# 310912; RRID:AB_314847 | 1:20 |

*Continued on next page*

*Continued*

| Reagent type (species) or resource | Designation | Source or reference | Identifiers | Additional information |
|---|---|---|---|---|
| Antibody | Anti-CD137-APC Fire 750 (Mouse monoclonal) | BioLegend | Cat# 309834; RRID:AB_2734280 | 1:20 |
| Antibody | Anti-CD8-PB (Mouse monoclonal) | BioLegend | Cat# 301023; RRID:AB_493110 | 1:50 |
| Antibody | Anti-CD3-BV510 (Mouse monoclonal) | BioLegend | Cat# 300447; RRID:AB_2563467 | 1:200 |
| Antibody | Anti-CD4-PerCP-Cy5.5 (Mouse monoclonal) | BD Biosciences | Cat# 332772; RRID:AB_2868621 | 1:25 |
| Antibody | Anti-CD8-APC (Mouse monoclonal) | BioLegend | Cat# 344722; RRID:AB_2075388 | 1:50 |
| Antibody | Anti-CD3-FITC (Mouse monoclonal) | E-bioscience | Cat# 11-0036-42 | 1:25 |
| Antibody | Anti-CD3-V500 (Mouse monoclonal) | BD Biosciences | Cat# 561416; RRID:AB_10612021 | 1:50 |
| Antibody | Anti-CD4-APC-H7 (Mouse monoclonal) | BD Biosciences | Cat# 641407; RRID:AB_1645733 | 1:50 |
| Antibody | Anti-CCR7-BV786 (Mouse monoclonal) | BD Biosciences | Cat# 564355; RRID:AB_2738765 | 1:30 |
| Antibody | Anti-CD45RA-BV605 (Mouse monoclonal) | BioLegend | Cat# 304133; RRID:AB_11126164 | 1:50 |
| Antibody | Anti-CTLA4-BV711 (Mouse monoclonal) | BioLegend | Cat# 369631; RRID:AB_2892450 | 1:50 |
| Antibody | Anti-CD160-PE-Cy7 (Mouse monoclonal) | BioLegend | Cat# 143009; RRID:AB_2562677 | 1:50 |
| Antibody | Anti-CD28-AF700 (Mouse monoclonal) | BioLegend | Cat# 302919; RRID:AB_528785 | 1:50 |
| Antibody | Anti-PD1-PE (Mouse monoclonal) | eBioscience | Cat# 12-9969-42 | 1:20 |
| Antibody | Anti-HLA-DR-FITC (Mouse monoclonal) | BD Biosciences | Cat# 347400; RRID:AB_2868846 | 1:60 |
| Antibody | Anti-CD38-PE (Mouse monoclonal) | BD Biosciences | Cat# 345806; RRID:AB_2868828 | 1:100 |
| Antibody | Anti-CD27-APCeFluor 780 (Mouse monoclonal) | eBioscience | Cat# 47-0271-82 | 1:20 |
| Antibody | Anti-CD28-PerCP Cy5.5 (Mouse monoclonal) | BD Biosciences | Cat# 337181; RRID:AB_513211 | 1:20 |
| Antibody | Anti-CD4-PE-Cy7 (Mouse monoclonal) | BD Biosciences | Cat# 348809; RRID:AB_2783789 | 1:250 |
| Antibody | Anti-CD57-APC (Mouse monoclonal) | BD Biosciences | Cat# 560845; RRID:AB_10563760 | 1:500 |
| Commercial assay or kit | DNA-free DNA Removal Kit | Thermo Fisher Scientific | Cat# AM1906 | |
| Chemical compound, drug | Platinum Quantitative PCR SuperMix-UDG | Thermo Fisher Scientific | Cat# 11730–025 | |
| Commercial assay or kit | TaqMan β-Actin Detection Reagents | Thermo Fisher Scientific | Cat# 401846 | |
| Commercial assay or kit | TaqMan Ribosomal RNA Control Reagents | Thermo Fisher Scientific | Cat# 4308329 | |
| Chemical compound, drug | SuperScript III reverse transcriptase | Thermo Fisher Scientific | Cat# 18080-085 | |

*Continued on next page*

*Continued*

| Reagent type (species) or resource | Designation | Source or reference | Identifiers | Additional information |
|---|---|---|---|---|
| Chemical compound, drug | Random primers | Thermo Fisher Scientific | Cat# 48190-011 | |
| Chemical compound, drug | RNaseOUT Recombinant Ribonuclease Inhibitor | Thermo Fisher Scientific | Cat# 10777-019 | |
| Chemical compound, drug | ddPCR Supermix for Probes (No dUTP) | Bio-Rad | Cat# 1863024 | |
| Commercial assay or kit | Puregene Cell Kit | QIAGEN | Cat# 158043 | |
| Commercial assay or kit | Human IFN-γ DuoSet ELISA | R&D Systems | DY285B | |
| Commercial assay or kit | CellTrace Violet Cell Proliferation Kit | Thermo Fisher Scientific | C34557 | |
| Commercial assay or kit | CD4+ T-Cell Isolation Kit, human | Miltenyi Biotec | 130-096-533 | Isolation of untouched CD4 T-cells |
| Other | HLA-A*0201 SLYNTVATLY APC-labeled MHC class I dextramers | Immudex | WB03338 | 50 tests |
| Other | HLA-B*0702 GPGHKARVL APC-labeled MHC class I dextramers | Immudex | WH03590 | 50 tests |
| Peptide, recombinant protein | HIV subtype B consensus Env peptide pool | NIH AIDS Reagent Program | HRP-12540 | 2 µg/mL |
| Peptide, recombinant protein | HIV subtype B consensus Gag peptide pool | NIH AIDS Reagent Program | HRP-12425 | 2 µg/mL |
| Peptide, recombinant protein | HIV subtype B consensus Nef peptide pool | NIH AIDS Reagent Program | HRP-12545 | 2 µg/mL |
| Peptide, recombinant protein | HIV subtype B consensus Pol peptide pool | NIH AIDS Reagent Program | ARP-6208 | 2 µg/mL |
| Software, algorithm | Prism 10.6.0 | GraphPad Software | https://www.graphpad.com/ RRID:SCR_002798 | Statistics |
| Software, algorithm | IBM SPSS Statistics 28.0.1.0 | IBM | https://www.ibm.com/ products/spss-statistics RRID:SCR_016479 | Statistics |
| Software, algorithm | QuantaSoft 1.7.4 | Bio-Rad | RRID:SCR_025696 | ddPCR data analysis |
| Software, algorithm | Rotor-Gene 2.3.5 | QIAGEN | RRID:SCR_015740 | qPCR data analysis |
| Software, algorithm | FlowJo 10.8.1 | Becton Dickinson | https://www.flowjo.com/ RRID:SCR_008520 | Flow cytometry data analysis |

## Sex as a biological variable

The NOVA study is open for inclusion to both males and females; however, in the current study, only samples from male participants were available. This reflects the epidemiology of AHI/early HIV infection in the Netherlands, which concerns mostly males. We do believe more females should be included in this research to be able to translate the findings to the entire population that is currently living with HIV.

## Study approval

The NOVA cohort is a multicenter, observational, prospective cohort that was initiated in 2015 and includes participants diagnosed with an AHI/early HIV infection (*Dijkstra et al., 2021*). The study was approved by the Medical Ethics Committee of the Amsterdam UMC (NL51613.018.14), and all study participants gave written informed consent.

## Study design

The study design of the NOVA study, including treatment regimen and follow-up visits, has been described elsewhere (*Dijkstra et al., 2021*). In short, people were included if they were 18 years or older and were diagnosed during AHI as defined by Fiebig stage I-IV. In case of a positive western blot, participants could only be included if they had a documented negative HIV screening test <6 months

before inclusion. After diagnosis, participants were referred to an HIV treatment center and started a four-drug regimen of emtricitabine/tenofovir 200/245 mg (FTC/TDF), dolutegravir 50 mg (DTG), darunavir 800 mg, and ritonavir 100 mg (DRV/r) as soon as possible (preferably within 24 hr). After 4 weeks, when baseline genotyping and viral mutations conferring possible drug resistance were known, DRV/r was discontinued. Participants could enroll in three groups based on the preparedness of individuals to undergo extensive sampling. Participants who accepted immediate treatment and follow-up but declined additional blood and tissue sampling were included in study group 1, of which only routine clinical care pVL and CD4 count measurements were collected. For groups 2 and 3, PBMC and semen were collected at study visits and cryopreserved. In group 3, in addition, GALT, lymph node biopsies, and CSF were collected. In both groups, leukapheresis was performed at weeks 24 and 156. The participants selected for the current analysis were in care at Amsterdam University Medical Center, Erasmus University Medical Center, or Radboud University Medical Center. Apart from pVL and CD4/CD8 measurements, all virological and immunological assessments were performed centrally at the Amsterdam University Medical Center.

## Viral load quantification and HIV subtyping

Viral load (HIV RNA) was measured in plasma using a sensitive HIV RNA assay. The assays that were used were m2000rt HIV RNA (Abbott) with a lower limit of quantification (LLOQ) of 40 copies/mL from 2015 to 2021, Alinity m HIV-1 Assay (Abbott) with an LLOQ of 20 copies/mL from 2021 onward (Amsterdam University Medical Center), COBAS AmpliPrep/COBAS TaqMan HIV-1 test (Roche Diagnostics), LLOQ 20 copies/mL and Aptima HIV-1 Quant Dx Assay (Hologic), LLOQ 30 copies/mL (Erasmus Medical Center), and Xpert HIV-1 assay (Cepheid) with an LLOQ of 40 copies/mL (Radboud University Medical Center). HIV subtypes were determined using neighbor-joining analysis to create phylogenetic trees. Reference sequences from the major HIV subtypes were obtained from the NCBI database, and the distance between sequences was calculated using the Kimura-2 parameter model.

## Quantification of total HIV DNA and cell-associated US HIV RNA

Total HIV DNA and cell-associated US HIV RNA were quantified by semi-nested qPCR according to the principles described previously (*Pasternak et al., 2008*). In brief, total nucleic acids were extracted from PBMCs using the Boom isolation method (*Boom et al., 1990*). Extracted cellular RNA was treated with DNase (DNA-free Kit; Thermo Fisher Scientific) to remove genomic DNA that could interfere with the quantitation and reverse-transcribed into cDNA using random primers and Super-Script III reverse transcriptase (all from Thermo Fisher Scientific). To quantify cell-associated US HIV RNA or total HIV DNA, this cDNA, or DNA extracted from PBMCs, respectively, was pre-amplified using primer pair $\Psi$_F (*Bruner et al., 2019*) and HIV-FOR (*Malnati et al., 2008*). The product of this PCR was used as template for a semi-nested qPCR with the $\Psi$ primer/probe combination (*Bruner et al., 2019*). HIV DNA or RNA copy numbers were determined using a 7-point standard curve with a linear range of more than 5 orders of magnitude that was included in every qPCR run and normalized to the total cellular DNA (by measurement of β-actin DNA) or RNA (by measurement of 18S ribosomal RNA) inputs, respectively, as described previously (*Pasternak et al., 2009*). Non-template control wells were included in every qPCR run and were consistently negative. Total HIV DNA and US RNA were detectable in 88.2% and 73.5% of the samples, respectively. Undetectable measurements of US RNA or total DNA were assigned to the values corresponding to 50% of the corresponding assay detection limits. The detection limits depended on the amounts of the normalizer (input cellular DNA or RNA) and therefore differed among samples. HIV transcription levels per provirus (US RNA/total DNA ratios) were calculated, taking into account that $10^6$ PBMCs contain 1 μg of total RNA (*Fischer et al., 1999*).

## Quantification of intact and defective HIV DNA

Intact and defective HIV DNA was quantified by the IPDA (*Bruner et al., 2019*). In brief, genomic DNA was isolated from PBMCs using Puregene Cell Kit (QIAGEN Benelux B.V.) according to the manufacturer's instructions and digested with *Bgl*I restriction enzyme (Thermo Fisher Scientific) as described previously (*Levy et al., 2021*). Notably, only a small minority (<8%) of HIV clade B sequences contain *Bgl*I recognition sites between $\Psi$ and env amplicons; therefore, *Bgl*I digestion is not expected to substantially influence the IPDA output, while improving the assay sensitivity by increasing the genomic

DNA input into a droplet digital PCR (ddPCR) (*Levy et al., 2021*). After desalting by ethanol precipitation, genomic DNA was subjected to two separate multiplex ddPCR assays: one targeting HIV Ψ and *env* regions using primers and probes described previously, including the unlabeled *env* competitor probe to exclude hypermutated sequences (*Bruner et al., 2019*), and the other targeting the cellular *RPP30* gene, which was measured to correct for DNA shearing and to normalize the intact HIV DNA to the cellular input. The *RPP30* assay amplified two regions, with amplicons located at exactly the same distance from each other as HIV Ψ and *env* amplicons. The first region was amplified using a forward primer 5'-AGATTTGGACCTGCGAGCG-3', a reverse primer 5'-GAGCGGCTGTCTCCAC AAGT-3', and a fluorescent probe 5'-FAM-TTCTGACCTGAAGGCTCTGCGCG-BHQ1-3' (*Luo et al., 2005*). The second region was amplified using a forward primer 5'-AGAGAGCAACTTCTTCAAGG G-3', a reverse primer 5'-TCATCTACAAAGTCAGAACATCAGA-3', and a fluorescent probe 5'-HEX-CCCGGCTCTATGATGTTGTTGCAGT-BHQ1-3'. The ddPCR conditions were as described previously (*Bruner et al., 2019*) with some minor amendments: we used 46 cycles of denaturation/annealing/extension and the annealing/extension temperature was 60°C. Intact HIV DNA was detectable in 97.0%, 3' defective HIV DNA in 76.5%, and 5' defective HIV DNA in 100% of the samples. QuantaSoft (version 1.7.4) was used for the data analysis. Positive and negative droplets were discriminated by manual thresholding.

## Quantitative viral outgrowth assay

Isolation of replication-competent virus was performed using CD4+ T-cell isolated PBMCs as described previously (*van 't Wout et al., 2008*). PBMCs were thawed and CD4+ T-cells were isolated by negative selection using MACS Microbeads (Miltenyi Biotec, Bergisch Gladbach, Germany). A median of $20.5\times10^6$ CD4+ T-cells (IQR 12.5–39) per sample was used for the QVOA. For the QVOA, CD4+ T-cells were prestimulated for 48 hr by anti-CD3 (immobilized) (1XE) and anti-CD28 (15E8, 3 mg/mL) in Iscove's modified Dulbecco's medium (IMDM) supplemented with 10% (vol/vol) heat-inactivated fetal calf serum (FCS), penicillin (100 U/mL), streptomycin (100 μg/mL), in a humidified 10% $CO_2$ incubator at 37°C. Subsequently, the CD4+ T-cells were co-cultured with 2-day PHA-stimulated donor PBMC in IMDM supplemented with 10% (vol/vol) heat-inactivated FCS, penicillin (100 U/mL), streptomycin (100 μg/mL), and IL-2 (20 U/mL; Chiron Benelux). Every 7 days, fresh PHA-stimulated donor PBMCs were added to propagate the culture. Culture supernatants were regularly analyzed for viral replication using an in-house p24 antigen enzyme-linked immunosorbent assay (ELISA) (*van 't Wout et al., 2008*).

## Lymphocyte count determination

CD4+ and CD8+ T-cell counts were determined using cytometry at the study sites.

## HLA typing

HLA genotyping was performed at the Department of Immunogenetics (Sanquin) by the PCR using sequence-based typing method (GenDx Products, Utrecht, the Netherlands) and real-time (RT)-PCR (Thermo Fisher, West Hills, CA, USA).

## Immune phenotyping of T-cells

PBMCs were used for immune phenotyping of CD8+ T-cells. T-cell activation was defined as the proportion of cells positive for CD38 and HLA-DR; naïve T-cells as the proportion of CD45RA and CD27 positive cells, memory T-cells as proportion of CD45RA negative and CD27 positive cells, and effector as proportion of CD45RA negative and CD27 negative cells. T-cell senescence was defined as CD27 and CD28 double negative cells. The following antibodies were used for staining: monoclonal antibody detecting CD3 (V500), CD4 (APC-H7), CCR7 (BV786) from BD Biosciences (San Jose, CA, USA); CD8 (Pacific Blue), CD45RA (BV605), CTLA4 (BV711), CD160 (PE-Cy7), CD28 (AF700) from BioLegend; and PD1 (PE) from eBioscience (San Diego, CA, USA), HLA-DR (FITC), CD38 (PE), CD27 (APCeFluor 780), CD28 (PerCP Cy5.5), CD4+ (PE-Cy7), CD57 (APC) from BD Biosciences (San Jose, CA, USA). The proportion of HIV-specific CD8+ T-cells was determined using APC-labeled MHC class I dextramers (Immudex, Virum, Denmark) carrying HLA-A*0201 SLYNTVATLY and HLA-B*0702 GPGH-KARVL molecules in combination with CD3 (V500) and CD4 (APC-H7) from BD and CD8 (Pacific Blue) from BioLegend.

Fluorescence was measured on the FACS Canto II (BD Biosciences). The fractions of cells expressing a marker alone or in combination or the mean fluorescence intensity were determined using FlowJo 10.8.1 (Becton Dickinson).

## Functional HIV cellular immune responses

An IFN-γ release assay was performed to evaluate the immune response upon HIV-peptide pool stimulation. A total of $0.5\times10^6$ PBMCs were stimulated with HIV consensus B Env, Gag, Nef, and Pol peptide pools (2 µg/mL, NIH AIDS Reagent Program) or cultured in medium alone as a control. After 1 day, culture supernatants were harvested, and IFN-γ released by the cells was determined by human IFN-γ DuoSet ELISA (R&D Systems, Minneapolis, MN, USA).

The AIM assay was performed to assess the frequency of reactive CD8+ T-cells. Therefore, PBMCs were stimulated for 6 hr with HIV-peptide pools (Env, Gag, Nef, and Pol) and then stained for flow cytometry. Reactive T-cells were determined by co-expression of CD137 (APC-H7/APC-Fire750) and CD69 (PE-Cy7) within the CD4+ and CD8+ T-cell populations, respectively. Fluorescence was measured on the FACS Canto II fluorescence-activated cell sorter (BD Biosciences). Marker expression levels were analyzed using FlowJo 10.8.1.

Proliferation of CD8+ T-cells upon antigen stimulation was assessed through the use of the Cell-Trace Violet Cell Proliferation Kit (Thermo Fisher). Cells were stained with CellTrace Violet according to the manufacturer's protocol (0.5 µM final concentration), and flow cytometry analysis was used to determine that all the cells were labeled with CellTrace Violet. Subsequently, the cells were stimulated with an HIV consensus B Gag, Env, Pol, and Nef peptide pool (2 µg/mL final concentration, NIH AIDS Reagent Program). An unstimulated control and positive controls using a peptide pool of CMVpp65 (2 µg/mL final concentration, NIH AIDS Reagent Program) or α-CD3 in combination with α-CD28 were included. After 7 days, cells were stained with FITC CD3, PerCP-Cy5.5 CD4 (BD Biosciences), and APC CD8 (BioLegend) for 30 min at 4°C. After fixation of the cells with CellFIX (BD), samples were analyzed on the BD FACSCanto II to assess the proliferation of CD8+ T-cells under the different conditions. The proportion of proliferating cells was determined using FlowJo 10.8.1. The precursor frequency is calculated as follows: per generation, the amount of CD8+ T-cells that proliferated was calculated (number of cells * $2^{(generation)}$); the precursor frequency is the total number of CD8+ T-cells that proliferated per 100 CD8+ T-cells (total CD8+ T-cells).

## Statistical analysis

Reservoir measurements and T-cell responses were compared between 24 and 156 weeks of ART using paired non-parametric Wilcoxon signed-rank tests. Strength of correlations between different reservoir measures and between reservoir measures and T-cell responses was tested using non-parametric Spearman correlation analyses, with Benjamini-Hochberg corrections for multiple comparisons (false discovery rate, 0.25). Intact proviral DNA levels were compared between participants with positive vs. negative QVOA using the Mann-Whitney test. Proportions of participants with intact proviral DNA higher than 100 copies/$10^6$ PBMC were compared between those with positive vs. negative QVOA using Fisher's exact test. Predictive value of variables measured at 24 weeks ART for the changes in HIV reservoir measures between 24 and 156 weeks of ART was modeled using generalized linear models (GLMs). GLMs were fitted on rank-transformed dependent variables, and results of type III tests are reported. Data were analyzed using Prism 10.6.0 (GraphPad Software) and IBM SPSS Statistics (version 28.0.1.0). All tests were two-sided. p-Values<0.05 were considered statistically significant.

## Acknowledgements

We would like to kindly thank all participants of the NOVA cohort study. We would also like to thank all medical doctors, lab personnel, and research nurses, including A Weijsenfeld and F Pijnappel (Amsterdam University Medical Center), A Karisli (Erasmus University Medical Center), and K Grintjes (Radboud University Medical Center). This research was funded by Gilead Sciences, funding number CO-NL-985-6195, and Aidsfonds, funding number P-60803. AOP acknowledges grant support from amfAR, The Foundation for AIDS Research (grant no. 1110680-77-RPRL), and from Partnership NWO-Dutch AIDS Fonds 'HIV cure for everyone' (grant no. KICH2.V4P.AF23.001).

# Additional information

## Competing interests

Casper Rokx: CR received research grants for investigator initiated studies, and reimbursement for scientific advisory board participation and travel from Gilead Sciences and ViiV Healthcare. Monique Nijhuis: MN received consultancy fees from ViiV Healthcare. The other authors declare that no competing interests exist.

## Funding

| Funder | Grant reference number | Author |
|---|---|---|
| Gilead Sciences | CO-NL-985-6195 | Godelieve J de Bree |
| Aids Fonds | P-60803 | Godelieve J de Bree |
| amfAR, The Foundation for AIDS Research | 1110680-77-RPRL | Alexander O Pasternak |
| Nederlandse Organisatie voor Wetenschappelijk Onderzoek | KICH2.V4P.AF23.001 | Alexander O Pasternak Cynthia Lungu Casper Rokx Jori Symons Monique Nijhuis Neeltje A Kootstra Godelieve J de Bree |

The funders had no role in study design, data collection, and interpretation, or the decision to submit the work for publication.

## Author contributions

Pien Margien van Paassen, Conceptualization, Formal analysis, Validation, Investigation, Visualization, Methodology, Writing – original draft, Project administration, Writing – review and editing; Alexander O Pasternak, Conceptualization, Resources, Formal analysis, Supervision, Funding acquisition, Investigation, Visualization, Methodology, Writing – original draft, Writing – review and editing; Dita C Bolluyt, Formal analysis, Investigation, Visualization, Writing – review and editing; Karel A van Dort, Ad C van Nuenen, Validation, Investigation, Methodology, Writing – review and editing; Irma Maurer, Brigitte Boeser-Nunnink, Validation, Investigation, Writing – review and editing; Ninée VEJ Buchholtz, Formal analysis, Investigation, Writing – review and editing; Tokameh Mahmoudi, Casper Rokx, Monique Nijhuis, Conceptualization, Resources, Writing – review and editing; Cynthia Lungu, Reinout van Crevel, Annelou LIP van der Veen, Liffert Vogt, Resources, Writing – review and editing; Jori Symons, Resources, Formal analysis, Methodology, Writing – review and editing; Michelle J Klouwens, Resources, Supervision, Writing – review and editing; Jan M Prins, Conceptualization, Resources, Supervision, Funding acquisition, Writing – review and editing; Neeltje A Kootstra, Godelieve J de Bree, Conceptualization, Resources, Supervision, Funding acquisition, Methodology, Writing – original draft, Project administration, Writing – review and editing

## Author ORCIDs

Pien Margien van Paassen ⓘ https://orcid.org/0000-0002-7300-3232
Alexander O Pasternak ⓘ https://orcid.org/0000-0002-4097-4251
Tokameh Mahmoudi ⓘ https://orcid.org/0000-0002-2060-9353
Neeltje A Kootstra ⓘ https://orcid.org/0000-0001-9429-7754
Godelieve J de Bree ⓘ https://orcid.org/0000-0002-5852-5833

## Ethics

The study was approved by the Medical Ethics Committee of the Amsterdam UMC (NL51613.018.14) and all study participants gave written informed consent.

Reviewer #2 (Public review): https://doi.org/10.7554/eLife.106402.3.sa1
Author response https://doi.org/10.7554/eLife.106402.3.sa2

## Additional files

### Supplementary files
MDAR checklist

### 1Data availability
Source data for all figures and tables is uploaded.

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
