## [Editor Report · eLife Assessment]

The findings of this study are **valuable** as it demonstrates that when treatment is initiated during acute infection, HIV specific CD8 T cell responses are maintained long term and continued proliferative capacity of these cells may play a role in reducing HIV DNA levels. The evidence supporting the conclusions are **solid** with rigorous and advanced methodology used with the major limitations being that the findings are association level and do not meet strict criteria for causality. The work is of interest to the HIV cure field and suggests that enhancing early HIV specific CD8 T cell responses should be considered in the design of interventional cure strategies.

---

## [Referee Report · Reviewer #2 (Public review)]

This study investigated the impact of early HIV specific CD8 T cell responses on the viral reservoir size after 24 weeks and 3 years of follow up in individuals who started ART during acute infection. Viral reservoir quantification showed that total and defective HIV DNA, but not intact, declined significantly between 24 weeks and 3 years post-ART. The authors also showed that functional HIV-specific CD8⁺ T-cell responses persisted over three years and that early CD8⁺ T-cell proliferative capacity was linked to reservoir decline, supporting early immune intervention in the design of curative strategies.

The paper is well written, easy to read, and the findings are clearly presented. The study is novel as it demonstrates the effect of HIV specific CD8 T cell responses on different states of the HIV reservoir, that is HIV-DNA (intact and defective), the transcriptionally active and inducible reservoir. Although small, the study cohort was relevant and well-characterized as it included individuals who initiated ART during acute infection, 12 of whom were followed longitudinally for 3 years, providing unique insights into the beneficial effects of early treatment on both immune responses and the viral reservoir. The study uses advanced methodology. I enjoyed reading the paper.

The study's limitations are minor and well acknowledged. While the cohort included only male participants-potentially limiting generalizability-the authors have clarified this limitation in the discussion. Although a chronic infection control group was not yet available, the authors explained that their protocol includes plans to add this comparison in future studies. These limitations are appropriately addressed and do not undermine the strength or validity of the study's conclusions.

---

## [Author Response]

The following is the authors’ response to the original reviews.

**Reviewer #1 (Public review):**
Summary:In this work, van Paassen et al. have studied how CD8 T cell functionality and levels predict HIV DNA decline. The article touches on interesting facets of HIV DNA decay, but ultimately comes across as somewhat hastily done and not convincing due to the major issues.(1) The use of only 2 time points to make many claims about longitudinal dynamics is not convincing. For instance, the fact that raw data do not show decay in intact, but do for defective/total, suggests that the present data is underpowered. The authors speculate that rising intact levels could be due to patients who have reservoirs with many proviruses with survival advantages, but this is not the parsimonious explanation vs the data simply being noisy without sufficient longitudinal follow-up. n=12 is fine, or even reasonably good for HIV reservoir studies, but to mitigate these issues would likely require more time points measured per person.(1b) Relatedly, the timing of the first time point (6 months) could be causing a number of issues because this is in the ballpark for when the HIV DNA decay decelerates, as shown by many papers. This unfortunate study design means some of these participants may already have stabilized HIV DNA levels, so earlier measurements would help to observe early kinetics, but also later measurements would be critical to be confident about stability.

The main goal of the present study was to understand the relationship of the HIV-specific CD8 T-cell responses early on ART with the reservoir changes across the subsequent 2.5-year period on suppressive therapy. We have revised the manuscript in order to clarify this. We chose these time points because the 24 week time point is past the initial steep decline of HIV DNA, which takes place in the first weeks after ART initiation. It is known that HIV DNA continues to decay for years after (Besson, Lalama et al. 2014, Gandhi, McMahon et al. 2017).

(2) Statistical analysis is frequently not sufficient for the claims being made, such that overinterpretation of the data is problematic in many places.(2a) First, though plausible that cd8s influence reservoir decay, much more rigorous statistical analysis would be needed to assert this directionality; this is an association, which could just as well be inverted (reservoir disappearance drives CD8 T cell disappearance).

To correlate different reservoir measures between themselves and with CD8+ T-cell responses at 24 and 156 weeks, we now performed non-parametric (Spearman) correlation analyses, as they do not require any assumptions about the normal distribution of the independent and dependent variables. Benjamini-Hochberg corrections for multiple comparisons (false discovery rate, 0.25) were included in the analyses and did not change the results.

Following this comment we would like to note that the association between the T-cell response at 24 weeks and the subsequent decrease in the reservoir cannot be bi-directional (that can only be the case when both variables are measured at the same time point). Therefore, to model the predictive value of T-cell responses measured at 24 weeks for the decrease in the reservoir between 24 and 156 weeks, we fitted generalized linear models (GLM), in which we included age and ART regimen, in addition to three different measures of HIV-specific CD8+ T-cell responses, as explanatory variables, and changes in total, intact, and total defective HIV DNA between 24 and 156 weeks ART as dependent variables.

(2b) Words like "strong" for correlations must be justified by correlation coefficients, and these heat maps indicate many comparisons were made, such that p-values must be corrected appropriately.

We have now used Spearman correlation analysis, provided correlation coefficients to justify the wording, and adjusted the p-values for multiple comparisons (Fig. 1, Fig 3., Table 2). Benjamini-Hochberg corrections for multiple comparisons (false discovery rate, 0.25) were included in the analyses and did not change the results.

(3) There is not enough introduction and references to put this work in the context of a large/mature field. The impacts of CD8s in HIV acute infection and HIV reservoirs are both deep fields with a lot of complexity.

Following this comment we have revised and expanded the introduction to put our work more in the context of the field (CD8s in acute HIV and HIV reservoirs).

**Reviewer #2 (Public review):**
Summary:This study investigated the impact of early HIV specific CD8 T cell responses on the viral reservoir size after 24 weeks and 3 years of follow-up in individuals who started ART during acute infection. Viral reservoir quantification showed that total and defective HIV DNA, but not intact, declined significantly between 24 weeks and 3 years post-ART. The authors also showed that functional HIV-specific CD8⁺ T-cell responses persisted over three years and that early CD8⁺ T-cell proliferative capacity was linked to reservoir decline, supporting early immune intervention in the design of curative strategies.Strengths:The paper is well written, easy to read, and the findings are clearly presented. The study is novel as it demonstrates the effect of HIV specific CD8 T cell responses on different states of the HIV reservoir, that is HIV-DNA (intact and defective), the transcriptionally active and inducible reservoir. Although small, the study cohort was relevant and well-characterized as it included individuals who initiated ART during acute infection, 12 of whom were followed longitudinally for 3 years, providing unique insights into the beneficial effects of early treatment on both immune responses and the viral reservoir. The study uses advanced methodology. I enjoyed reading the paper.Weaknesses:All participants were male (acknowledged by the authors), potentially reducing the generalizability of the findings to broader populations. A control group receiving ART during chronic infection would have been an interesting comparison.

We thank the reviewer for their appreciation of our study. Although we had indeed acknowledged the fact that all participants were male, we have clarified why this is a limitation of the study (Discussion, lines 296-298). The reviewer raises the point that it would be useful to compare our data to a control group. Unfortunately, these samples are not yet available, but our study protocol allows for a control group (chronic infection) to ensure we can include a control group in the future.

**Reviewer #1 (Recommendations for the authors):**
Minor:On the introduction:(1) One large topic that is mostly missing completely is the emerging evidence of selection on HIV proviruses during ART from the groups of Xu Yu and Matthias Lichterfeld, and Ya Chi Ho, among others.

Previously, it was only touched upon in the Discussion. Now we have also included this in the Introduction (lines 77-80).

(2) References 4 and 5 don't quite match with the statement here about reservoir seeding; we don't completely understand this process, and certainly, the tissue seeding aspect is not known.

Line 61-62: references were changed and this paragraph was rewritten to clarify.

(3) Shelton et al. showed a strong relationship with HIV DNA size and timing of ART initiation across many studies. I believe Ananwaronich also has several key papers on this topic.

References by Ananwaronich are included (lines 91-94).

(4) "the viral levels decline within weeks of AHI", this is imprecise, there is a peak and a decline, and an equilibrium.

We agree and have rewritten the paragraph accordingly.

(5) The impact of CD8 cells on viral evolution during primary infection is complex and likely not relevant for this paper.

We have left viral evolution out of the introduction in order to keep a focus on the current subject.

(6) The term "reservoir" is somewhat polarizing, so it might be worth mentioning somewhere exactly what you think the reservoir is, I think, as written, your definition is any HIV DNA in a person on ART?

Indeed, we refer to the reservoir when we talk about the several aspects of the reservoir that we have quantified with our assays (total HIV DNA, unspliced RNA, intact and defective proviral DNA, and replication-competent virus). In most instances we try to specify which measurement we are referring to. We have added additional reservoir explanation to clarify our definition to the introduction (lines 55-58).

(7) I think US might be used before it is defined.

We thank the reviewer for this notification, we have now also defined it in the Results section (line 131).

(8) In Figure 1 it's also not clear how statistics were done to deal with undetectable values, which can be tricky but important.

We have now clarified this in the legend to Figure 2 (former Figure 1). Paired Wilcoxon tests were performed to test the significance of the differences between the time points. Pairs where both values were undetectable were always excluded from the analysis. Pairs where one value was undetectable and its detection limit was higher than the value of the detectable partner, were also excluded from the analysis. Pairs where one value was undetectable and its detection limit was lower than the value of the detectable partner, were retained in the analysis.

In the discussion:(1) "This confirms that the existence of a replication-competent viral reservoir is linked to the presence of intact HIV DNA." I think this statement is indicative of many of the overinterpretations without statistical justification. There are 4 of 12 individuals with QVOA+ detectable proviruses, which means there are 8 without. What are their intact HIV DNA levels?

We thank the reviewer for the question that is raised here. We have now compared the intact DNA levels (measured by IPDA) between participants with positive vs. negative QVOA output, and observed a significant difference. We rephrased the wording as follows: “We compared the intact HIV DNA levels at the 24-week timepoint between the six participants, from whom we were able to isolate replicating virus, and the fourteen participants, from whom we could not. Participants with positive QVOA had significantly higher intact HIV DNA levels than those with negative QVOA (p=0.029, Mann-Whitney test; Suppl. Fig. 3). Five of six participants with positive QVOA had intact DNA levels above 100 copies/106 PBMC, while thirteen of fourteen participants with negative QVOA had intact HIV DNA below 100 copies/106 PBMC (p=0.0022, Fisher’s exact test). These findings indicate that recovery of replication-competent virus by QVOA is more likely in individuals with higher levels of intact HIV DNA in IPDA, reaffirming a link between the two measurements.”

(2) "To determine whether early HIV-specific CD8+ T-cell responses at 24 weeks were predictive for the change in reservoir size". This is a fundamental miss on correlation vs causation... it could be the inverse.

We thank the reviewer for the remark. We have calculated the change in reservoir size (the difference between the reservoir size at 24 weeks and 156 weeks ART) and analyzed if the HIVspecific CD8+ T-cell response at 24 weeks ART are predictive for this change. We do not think it can be inverse, as we have a chronological relationship (CD8+ responses at week 24 predict the subsequent change in the reservoir).

(3) "This may suggest that active viral replication drives the CD8+ T-cell response." I think to be precise, you mean viral transcription drives CD8s, we don't know about the full replication cycle from these data.

We agree with the reviewer and have changed “replication” to “transcription” (line 280).

(4) "Remarkably, we observed that the defective HIV DNA levels declined significantly between 24 weeks and 3 years on ART. This is in contrast to previous observations in chronic HIV infection (30)". I don't find this remarkable or in contrast: many studies have analyzed and/or modeled defective HIV DNA decay, most of which have shown some negative slope to defective HIV DNA, especially within the first year of ART. See White et al., Blankson et al., Golob et al., Besson et al., etc In addition, do you mean in long-term suppressed?

The point we would like to make is that, compared to other studies, we found a significant, prominent decrease in defective DNA (and not intact DNA) over the course of 3 years, which is in contrast to other studies (where usually the decrease in intact is significant and the decrease in defective less prominent). We have rephrased the wording (lines 227-230) as follows:

“We observed that the defective HIV DNA levels decreased significantly between 24 and 156 weeks of ART. This is different from studies in CHI, where no significant decrease during the first 7 years of ART (Peluso, Bacchetti et al. 2020, Gandhi, Cyktor et al. 2021), or only a significant decrease during the first 8 weeks on ART, but not in the 8 years thereafter, was observed (Nühn, Bosman et al. 2025).”

**Reviewer #2 (Recommendations for the authors):**
(1) Page 4, paragraph 2 - will be informative to report the statistics here.(2) Page 4, paragraph 4 - "General phenotyping of CD4+ (Suppl. Fig. 3A) and CD8+ (Supplementary Figure 3B) T-cells showed no difference in frequencies of naïve, memory or effector CD8+ T-cells between 24 and 156 weeks." - What did the CD4+ phenotyping show?

We thank the reviewer for the remark. Indeed, there were also no differences in frequencies of naïve, memory or effector CD4+ T-cells between 24 and 156 weeks. We have added this to the paragraph (now Suppl. Fig 4), lines 166-168.

(3) Page 5, paragraph 3 - "Similarly, a broad HIV-specific CD8+ T-cell proliferative response to at least three different viral proteins was observed in the majority of individuals at both time points" - should specify n=? for the majority of individuals.

At time point 24 weeks, 6/11 individuals had a response to env, 10/11 to gag, 5/11 to nef, and 4/11 to pol. At 156 weeks, 8/11 to env, 10/11 to gag, 8/11 to nef and 9/11 to pol. We have added this to the text (lines 188-191).

(4) Seven of 22 participants had non-subtype B infection. Can the authors explain the use of the IPDA designed by Bruner et. al. for subtype B HIV, and how this may have affected the quantification in these participants?

Intact HIV DNA was detectable in all 22 participants. We cannot completely exclude influence of primer/probe-template mismatches on the quantification results, however such mismatches could also have occurred in subtype B participants, and droplet digital PCR that IPDA is based on is generally much less sensitive to these mismatches than qPCR.

(5) Page 7, paragraph 2 - the authors report a difference in findings from a previous study ("a decline in CD8 T cell responses over 2 years" - reference 21), but only provide an explanation for this on page 9. The authors should consider moving the explanation to this paragraph for easier understanding.

We agree with the reviewer that this causes confusion. Therefore, we have revised and changed the order in the Discussion.

(6) Page 7, paragraph 2 - Following from above, the previous study (21) reported this contradicting finding "a decline in CD8 T cell responses over 2 years" in a CHI (chronic HIV) treated cohort. The current study was in an acute HIV treated cohort. The authors should explain whether this may also have resulted in the different findings, in addition to the use of different readouts in each study.

We thank the reviewer for this attentiveness. Indeed, the study by Takata et al. investigates the reservoir and HIV-specific CD8+ T-cell responses in both the RV254/ SEARCH010 study who initiated ART during AHI and the RV304/ SEARCH013 who initiated ART during CHI. We had not realized that the findings of the decline in CD8 T cell responses were solely found in the RV304/ SEARCH013 (CHI cohort). It appears functional HIV specific immune responses were only measured in AHI at 96 weeks, so we have clarified this in the Discussion.

Besson, G. J., C. M. Lalama, R. J. Bosch, R. T. Gandhi, M. A. Bedison, E. Aga, S. A. Riddler, D. K. McMahon, F. Hong and J. W. Mellors (2014). "HIV-1 DNA decay dynamics in blood during more than a decade of suppressive antiretroviral therapy." Clin Infect Dis 59(9): 1312-1321.

Gandhi, R. T., J. C. Cyktor, R. J. Bosch, H. Mar, G. M. Laird, A. Martin, A. C. Collier, S. A. Riddler, B. J. Macatangay, C. R. Rinaldo, J. J. Eron, J. D. Siliciano, D. K. McMahon and J. W. Mellors (2021). "Selective Decay of Intact HIV-1 Proviral DNA on Antiretroviral Therapy." J Infect Dis 223(2): 225-233.

Gandhi, R. T., D. K. McMahon, R. J. Bosch, C. M. Lalama, J. C. Cyktor, B. J. Macatangay, C. R. Rinaldo, S. A. Riddler, E. Hogg, C. Godfrey, A. C. Collier, J. J. Eron and J. W. Mellors (2017). "Levels of HIV-1 persistence on antiretroviral therapy are not associated with markers of inflammation or activation." PLoS Pathog 13(4): e1006285.

Nühn, M. M., K. Bosman, T. Huisman, W. H. A. Staring, L. Gharu, D. De Jong, T. M. De Kort, N. Buchholtz, K. Tesselaar, A. Pandit, J. Arends, S. A. Otto, E. Lucio De Esesarte, A. I. M. Hoepelman, R. J. De Boer, J. Symons, J. A. M. Borghans, A. M. J. Wensing and M. Nijhuis (2025). "Selective decline of intact HIV reservoirs during the first decade of ART followed by stabilization in memory T cell subsets." Aids 39(7): 798-811.

Peluso, M. J., P. Bacchetti, K. D. Ritter, S. Beg, J. Lai, J. N. Martin, P. W. Hunt, T. J. Henrich, J. D. Siliciano, R. F. Siliciano, G. M. Laird and S. G. Deeks (2020). "Differential decay of intact and defective proviral DNA in HIV-1-infected individuals on suppressive antiretroviral therapy." JCI Insight 5(4).